# Susceptibility to Heart Defects in Down Syndrome Is Associated with Single Nucleotide Polymorphisms in HAS 21 Interferon Receptor Cluster and *VEGFA* Genes

**DOI:** 10.3390/genes11121428

**Published:** 2020-11-28

**Authors:** Carmela Rita Balistreri, Claudia Leonarda Ammoscato, Letizia Scola, Tiziana Fragapane, Rosa Maria Giarratana, Domenico Lio, Maria Piccione

**Affiliations:** 1Clinical Pathology, Department of Biomedicine, Neuroscience and Advanced Diagnostics (Bi.N.D.), University of Palermo, 90100 Palermo, Italy; carmelarita.balistreri@unipa.it (C.R.B.); letizia.scola@unipa.it (L.S.); rosamaria.giarratana@unipa.it (R.M.G.); 2Regional Reference Center for the Control and Treatment of Down Syndrome and Other Chromosomal and Genetic Diseases, Medical Genetics University of Palermo, Villa Sofia/Cervello-United Hospitals, 90100 Palermo, Italy; claudia.ammoscato@gmail.com (C.L.A.); maria.piccione@unipa.it (M.P.); 3Department of Neonatology, University Hospital Galway, h91 Galway, Ireland; tiziana.frapapane@hse.ie

**Keywords:** Down syndrome, heart defects, SNPs, IFNR, VEGFA

## Abstract

Background: Congenital heart defects (CHDs) are present in about 40–60% of newborns with Down syndrome (DS). Patients with DS can also develop acquired cardiac disorders. Mouse models suggest that a critical 3.7 Mb region located on human chromosome 21 (HSA21) could explain the association with CHDs. This region includes a cluster of genes (*IFNAR1, IFNAR2, IFNGR2, IL10RB*) encoding for interferon receptors (IFN-Rs). Other genes located on different chromosomes, such as the vascular endothelial growth factor A (*VEGFA*), have been shown to be involved in cardiac defects. So, we investigated the association between single nucleotide polymorphisms (SNPs) in *IFNAR2*, *IFNGR2*, *IL10RB* and *VEGFA* genes, and the presence of CHDs or acquired cardiac defects in patients with DS. Methods: Individuals (*n* = 102) with DS, and age- and gender-matched controls (*n* = 96), were genotyped for four SNPs (rs2229207, rs2834213, rs2834167 and rs3025039) using KASPar assays. Results: We found that the *IFNGR2* rs2834213 G homozygous genotype and *IL10RB* rs2834167G-positive genotypes were more common in patients with DSand significantly associated with heart disorders, while *VEGFA* rs3025039T-positive genotypes (T/*) were less prevalent in patients with CHDs. Conclusions: We identified some candidate risk SNPs for CHDs and acquired heart defects in DS. Our data suggest that a complex architecture of risk alleles with interplay effects may contribute to the high variability of DS phenotypes.

## 1. Introduction

Down syndrome (DS) is characterized by a high variability of clinical pictures and symptomatology that can involve many organs and tissues, in particular, the circulatory system, the central nervous system and the immune system. Such comorbidity in one or more organs and/or systems has been demonstrated to reduce the life expectancy of patients, with the development of phenotypic traits similar to those observed during aging in chromosomally normal subjects [1]. 

Consistent with these observations, the incidence of congenital heart defects (CHDs) is 0.8% in the general population and rises to approximately 40–60% in DS [2,3]. One of the largest studies in patients with DS confirmed this percentage, detecting cardiovascular abnormalities in 342 out of 821 (42%) of children with DS born between 1985 and 2006 in the north-eastern region of England [4]. A similar incidence rate (43.9%) has been recently reported by Santoro and coworkers in 230 Italian patients with DS [5]. Atrioventricular septal defects (AVSDs) are the most common forms of CHDs in DS, followed by ventricular septal defects (VSDs) and tetralogy of Fallot (TF) [6,7]. Moreover, some asymptomatic adolescents and adults without CHDs at birth may develop acquired valve anomalies and electric defects [8]. Despite the high incidence of CHDs in DS, the pathophysiology and the pathways behind them remain unknown, as well as the close relationship with DS.Gene mapping analysis studies on human chromosome 21 (HSA21) have revealed that the supernumerary genes related to DS onset (such as superoxide dismutase Cu/Zn, transcription factors, stress inducible factors, cytokine receptors and amyloid precursor protein) that codify molecules witha key role in different metabolic pathways involved in diverse processes, such as angiogenesis, as well as maintaining cardiovascular homeostasis [9]. Even though the genetic imbalance caused by the triplication of HSA21 can surely contribute to the impact on thepathways involved in heart development, it cannot be sufficient to determine CHD onset [10]. Thishas led to advancing various hypotheses on the genotype–phenotype relationship of DS with CHDs. The gene dosage imbalance hypothesis states that the increased copy number of genes that are on HSA21 can lead to an overall increase in gene and protein expression, resulting in the phenotypic manifestations associated with DS [11].

The amplified developmental instability hypothesis suggests that gene over-dosages on HSA21 result in a general disruption of genomic regulation and function of genes, inducing the expressionof the traits associated with DS [12]. The most accredited hypothesis, known as the critical region hypothesis, affirms that a small set of genes within the Down syndrome critical or chromosomal region (DSCR) is responsible for the development of common DS phenotypes [13,14].

However, data from animal models, such as mouse models, have contributed to evidence that CHDs in DS are likely associated with a cluster of genes. Variants of these genes (i.e., mutations and polymorphisms) may influence onset, phenotypic and severity heterogeneity of CHDs in DS. Liu et al., using the Ts1Cje mouse model for DS, identified a 3.7 Mb critical genomic region associated with DS heart defects [15,16]. This region is located between the *Tiam1* and *IL10RB* loci and contains a cluster of four interferon receptor (IFNR) genes: *IFNAR1*, *IFNAR2*, *IFNGR2* and *IL10RB*. *IFNAR1* and *IFNAR2* are implied in the expression of type I interferon (IFN-α/β) receptors. The IFN-γR2 chain is a component of the type II interferon receptor complex (IFN-γ) and IL-10Rβ participates in type III interferon(IFN-λ) receptor structure [17]. These data might lead to the hypothesis that mutation and common polymorphisms (single nucleotide polymorphisms, SNPs) of gene, such as IFN receptor gene SNP, mapping at HSA21 might be responsible for the heterogeneity of expression of the diverse organ/system anomalies characterizing the Down syndrome phenotypes, such as CHDs. Increased expression of these receptors in patients with DS has been demonstrated to contribute to autoimmune diseases [18]. Impaired activation of IFN signaling has also been reported in individuals with DS and periodontitis [19]. To our knowledge, at this moment, no data are available on the role of IFNreceptor cluster genetic assetsin heart defect pathogenesis. On the other hand, in animal models, the increased dosage of genes on the homolog of HSA21 explains only inpart the increased risk to CHDs and suggests the involvement of other genes located on other chromosomes [20,21,22]. A study, conducted on strains of mice that were trisomic for segments of MMU16, homologous to HSA21, demonstratedan overexpression of vascular endothelial growth factor (VEGF) in severe DS heart abnormalities, including an attenuated compact layer of myocardium, the overproduction of trabeculae, defective ventricular septation and remodeling of the outflow tract [20]. Data obtained in knockout VEGFreceptor mice have therefore showna functional and structural defect in cardiac valve development, thereby suggesting an involvement of these mediators in heart valve formation [21]. Furthermore, a recent meta-analysis reported a significant association of the rs1570360 andrs3025039 *VEGF-A* SNPs with the risk of CHD [23]. In addition, an excess of deleterious variants in VEGF-A pathway genes in DS has been associated with Atrial septum defects ASDs [24]. 

Based on these observations, we analyzed the frequencies of *IFNAR2* rs2229207, *IFNGR2* rs2834213, *IL10RB* rs2834167 and *VEGFA* rs3025039 SNPs in a group of young adults withDS compared to a group of healthy subjects, matched for age and gender, in order to evaluate whether these polymorphisms can influence the susceptibility of patients with DS to either CHDs or acquired defects.

## 2. Materials and Methods

### 2.1. Subjects

A group of 102 patients with DS to free HSA21trisomy, aged > 18 years, was recruited and clinically characterized at the “Regional Reference Center for the control and treatment of Down syndrome and other chromosomal and genetic diseases”, UOSD of Medical Genetics of the Villa Sofia-Cervello-United Hospitals from January 2019 to January 2020.Of the patients, 57.84% were affected by congenital and/or acquired cardiac pathologies (38.23% congenital cardiac pathologies and 19.61% acquired without previous congenital cardiac pathologies). A group of 96 healthy subjects matched for age and gender were recruited among the staff and students of the Clinical Pathology laboratory of the Department of Biomedicine, Neuroscience and Advanced Diagnostics of the University of Palermo (Table 1).

Our study was performed in accordance with ethical standards of the Helsinki Declaration of the World Medical Association and Italian legislation, and it received approval from the Regional Ethics Board in Palermo (Palermo 2 Ethical board registry nr. 260 AOR 2018). The participants or their legal guardiangave signed written informed consent. Data were encrypted to ensure patients’ and controls’ privacy.

### 2.2. Sample Collection Genotyping

Venous blood samples (16 mL) from all subjects were collected in tubes containing EDTA anticoagulant. Blood samples were stored at −80 degrees celsius until they were processedand DNA was derived from blood samples from all the enrolled individuals by a salting out procedure. DNA samples were typed, by using a procedure previously described [25], for the polymorphisms of the four selected candidate genes, as reported in Table 2. Information about these polymorphisms was acquired from dbSNP NCBI, the ENSEMBL database (http://www.ensembl.org/index.html), and the UCSC Genome Browser website (http://genome.ucsc.edu).The allelic and genotypic frequencies of these gene variants were detected using the assays on demand developed by KBioscience Ltd. (Middlesex, UK) and based on a homogeneous fluorescence resonance energy transfer (FRET) detection and allele-specific PCR (KASPar). Briefly, two specific oligonucleotides were designed for each allele of the SNPs studied. Each one of these oligos was tailed with 18 bp sequences distinct from each other. Taq polymerase, dNTPs, an internal standard dye (rhodamine X, Rox) and reverse primers were included. In addition, the KBioscience modified versions of Taq polymerase are unable to extend primers characterized to be mismatched at their 3′ terminal base. This property was used to discriminate the two alleles. The reaction was monitored by the fluorescence signals released by two other FRET reporter oligos included in the reaction mixes. The endpoint fluorescence emission was detected on an ABI-Prism 7300 Real-Time PCR Analyzer (Applied Biosystem, Carlsbad, CA, USA). The genotypes were determined using the 7300 system SDS software, version 1.3 (Applied Biosystems), sample by sample, based on the detection of unique (homozygous samples) or double (heterozygous samples) fluorescence signals. Typing of homozygous genotypes of SNPs located on HSA21 identified subjects bearing three copies of a single allele;heterozygosity was defined both by genotypes constituted by two major and one minor allele and bygenotypesconstituted by one major and two minor alleles.

### 2.3. Statistics

Allele and genotype frequencies were evaluated by gene count. Data were tested for goodness of fit between observed and expected genotype frequencies according to the Hardy–Weinberg equilibrium, by Pearson’s distribution and chi-square tests. Significant differences in genotype distributions among groups were calculated by using Fisher’s exact test. Multiple logistic regression models were applied using dominant (major allele homozygotes versus heterozygotes plus minor allele homozygotes), codominant (heterozygotes versus major allele homozygotes plus minor allele homozygotes) and recessive (major allele homozygotes plus heterozygotes versus minor allele homozygotes) models. Odds ratios (OR), 95% confidence intervals (95% C.I.) and *p*-values were determined using GraphPadInStat software version 3.06 (GraphPad, San Diego, CA, USA). A *p*-value< 0.05 was considered statistically significant.

## 3. Results

As reported in Table 3, *IFNAR2* rs2229207 genotypes showed similar frequencies in DS and controls and were not associated with susceptibility to cardiac pathologies in DS (Table 4).

Conversely, the *IFNGR2* rs2834213 GGG genotype, tagged as GG genotype, was more represented inpatients with DS than in controls (Table 5). In addition, the rs2834213 GG genotype was significantly more represented in patients with DS affected by congenital cardiac pathologiesthan in controls (Table 6). However, the distribution of *IFNGR2* rs2834213 genotype frequencies of the controls but not patientswith DS respected the Hardy–Weinberg equilibrium. On the other hand, statistical analyses reported in Table 6 demonstrate that the DS intra-group analyses showed that this genotype was more represented in patients with DS compared to patients without cardiac pathologies and appeared significantly over-represented in patients with DS with CHDs compared to those that developed cardiac damage after birth. In contrast, the *IFNGR2* rs2834213-positive genotypes (AAA, AAG and AGG, indicated as A/*) and in particular the heterozygous genotypes, tagged as AG, appear to be significantly under-represented in the group of patients with congenital lesions. These results suggested that this SNP might be directly or through interaction with other genes implied in the susceptibility to congenital heartdamage in patients with DS.

The *IL10RB* rs2834167 genotype distribution of controlsbut not patientswith DS respected the Hardy–Weinberg equilibrium. The homozygous allele A genotype frequency of rs2834167 was lower in patients with DS than in the control group (Table 7). In addition, as reported in Table 8, the frequency of *IL10RB* rs2834167 G-positive genotypes (GGG, GGA and GAA, tagged as G/*), in particular heterozygous genotypes, was significantly increased in patients with DS. Moreover, the intragroup analyses evidenced a significant reduction of homozygousgenotypesand a significant increase in heterozygous genotypes in patients with cardiac pathologies compared both to the controls and CHD-unaffected individuals with DS (Table 8). No significant differences were observed by stratifying results according to congenital or acquired cardiac pathologies.

Table 9 reports *VEGFA* rs3025039 genotype frequencies of subjects affected by DS, controls and DS subgroups classified according the presence of congenitaland/or acquired cardiac pathologies. As reported in Table 10, analyses of *VEGFA* rs3025039 genotype distributions demonstrated a significantly increased frequency of the CC genotype in the patient groupwith respect to control subjects. No significant differences were found comparing the two subgroups of patients with or without cardiac pathologies. On the other hand, statistical analyses demonstrated that the CT genotype frequency among Down syndrome patients with congenital cardiac pathologies is significantly lower when compared to the CT frequency of patients with heart defects developed after birth (AwoCC).

Comprehensively, our results indicate that susceptibility to CHDs in patients with DS is influenced by common genetic variation (SNPs) of genes coding for IFN receptor mapping on critical regions of HSA21 and by genes located on other chromosomes, such as VEGFA.

## 4. Discussion

As is well known, the detection of heart defects, in particular congenital defects (CHDs), is a common finding in patients with DS. [26]. Data from animal models indicate that a cluster of IFN receptor genes are closely related to a critical region located on chromosome 21 [9,15,16].

We selected *INFAR2* rs2229207T>C, *INFGR2* rs2834213>G and *IL10RB* rs2834167A>G SNPs to evaluate their relationship with the presence of congenital or acquired heart defects. The *INFAR2* rs2229207T>C SNP changes phenylalanine to serine at amino acid position 8 (F8S), inducing an increased response to IFNα and β [27]. The intronic (intron 2) *INFGR2* rs2834213>G SNP is not located near a splice site (5582 nucleotides downstream of a splice donor site and 877 nucleotides upstream of a splice acceptor site) but, on the other hand, is in linkage disequilibrium withthree SNPs contained in a 300 bp promoter segment, forming a haplotype associated with high transcriptional activity in vitro [28].

Our results evidenced that *IFNAR2* rs2229207 was not associated with significant susceptibility to cardiac pathologies in DS. Analyses of *IFNGR2* rs2834213 genotype distribution in the DS group showed a significant reduction of heterozygous genotypes compared to the expected frequency according to the Hardy–Weinberg (HW) equilibrium. Moreover, the G homozygous genotype was present in at least 30% of patients with DS and particularly in those with cardiac defects (Table 4). Stratifying genotype frequencies of patients with cardiac defects according to the presence of congenital or of acquired without previous congenital cardiac pathologies, the G homozygous genotype was significantly increased in the first group. Our data suggested that the presence of an over-dosage of the G allele may be implied in the susceptibility to heart congenital damage in patients with DS.

Sustained type I and II IFN responses were documented in different cell lines derived from DS tissues, as fibroblasts, Epstain-Barr Virus-transformed B cells, monocytes and T cells [29,30,31,32].However, as was recently reported, the IFN activation pathway appears not to be essential for heart development [33,34,35]. On the other hand, the activation of theses pathways, as a consequence ofinflammatory responses to intrauterine infections, induces interferon type I and type II production [36,37,38] and activation of the JAK/STAT pathway, which, via its target gene p21, might directly inhibit the proliferation and selection of Sca1-positive cardiac stem cells with a lack of differentiation of cardiomyocyte progenitors. In addition, human fetal cardiac stem cells exposed to high levels of IFN-γ acquire immune competences that are conserved in adults and in particular conditions might predispose to acquired heart damage [38,39]. For example, an adult fibroblast cell line from patients with DS exposed to IFN-γ has been demonstrated to slowin proliferation when compared to normal fibroblasts [29]. In addition, as recently stated by Qin et al. [40], interferon-γ inhibits the differentiation of stem cells from different types of tissue by inhibiting the activation of Notch signaling. Moreover, type I inflammation-mediated dysregulation of Notch4 signaling is involved in endothelial dysfunction [41]. Intriguingly, NOTCH4 is expressed in the developing heart and has previously been identified as a major player in early artery and endothelial-to-mesenchymal transformation, which is critical for endocardial cushiondifferentiation [42,43]. So, the *IFNGR2* rs2834213 G homozygous genotype might tag a genetically determined hyper-responsiveness to type II IFN, which in patients with DS, might be implied in susceptibility to CHDs by influencing crucial regulatory pathways in heart organogenesis.

Analyses of *IL10RB* rs2834167 genotype distribution in the Down syndrome patient group showed a significant reduction of homozygous A genotype compared to the expected HW equilibrium frequency. The G-positive (homozygous and heterozygous) genotypes were present in more than 60% of patients with DS and, particularly, in patients with cardiac defects (Table 7 and Table 8). Intragroup analyses demonstrated a significant reduction of the AA genotype and a significant increase in AG genotypes in patients with DS with cardiac pathologies with respect to patients without cardiac defects. In this subgroup of patients, the odds ratio of association with the G-positive genotypes was very high (OR: 9.714 C.I.: 3.605–26.17). The rs2834167A>G SNP that inducesa lysine to glutamic acid changein the *IL10RB* amino acidic sequence (K47E) seems to be associated with higher mRNA and cell-surface expression of IL-10Rβ [44]. Increased IL10RB expression have been found to be associated with cardiovascular diseases in older heart patients withstroke and in patients with DS affected by periodontitis [45,46,47].

IL10 RB engagement is the crucial event in IL-10/IL-10 receptor interaction to obtain the active receptor complex [48]. The ligand–receptor interaction activates via IL-10Rα Janus kinase-1 and tyrosine kinase-2, which induces the activation of tyrosine phosphorylation and latent transcription factor-3 (STAT3). Alternatively, IL-10 receptor β engagement, that down regulates STAT3 pathways, is involved in the activation pathway involving p-AKT and p-mTOR [47]. Cross talk between STAT3, AKT and mTOR modulates IL10-mediated inhibition of autophagy, one of the crucial mechanisms involved in cell development control and the clearance of damaged mitochondria by mitophagy [49]. mTOR is hyper-activated in DS, leading to mitochondrial dysfunction in intra- and extra-uterine life [50]. The increased levels of mTOR are actually implied in inhibition of autophagy induction, autophagosome formation and mitophagy [49]. Mitochondrial dysfunction hasrecently been strictly associatedwith aberrant ciliogenesis [51,52]. Cilia are emerging as an important hub for signaling processes crucial during development that are also needed for heart formation [52,53,54,55]. In this view, a genetically determined IL10RB hyper-dosage might play a central role in the mTOR-mediated dysregulation of mitochondrial homeostasis and organogenesis in DS.

On the other hand, it is reasonably likely that both *IFNGR2* rs2834213 and *IL10RB* rs2834167 risk genotypes represent the tags of other genetic variants on HSA21 relevant for CHDs. Actually, it is extremely improbable that a single polymorphism might be the principal agent responsible for susceptibility to CHDs in DS. Although the three copies ofthe genes of chromosome 21 certainly contribute to this risk, the increased gene dosage itself is not sufficient to cause pathologies, such as atrioventricular septal defects, that are extremely frequent in these patients. Indeed, half or more of those with DS have a structurally normal heart. Clearly, other factors in addition to trisomy 21 are required to cause CHDs in children with DS. However, the functional effects of *IFNGR2* rs2834213 and *IL10RB* rs2834167 SNPs might be directly involved in alteration of the signaling cascade that in the presence of other critical genetic variants might lead to CHD development.

A recent paper [22] suggests that rare variants in the Notch pathway and ciliome genes are associated with AVSDs in DS. These results provide further support for the involvement of the Notch pathway. In this view, our results allow us to hypothesize that minor genetic variants of the interferon receptor gene cluster (i.e., *IFNGR2* rs2834213 G homozygous genotype and *IL10RB* rs2834167 G positive genotypes) on HSA21 might be considered as tags ofa complex genetic constellation involving over-dosage of genetic susceptibility variantsonHSA21 and the presence of other susceptibility alleles on other chromosomes, resultingin CHDsfrequently, but not exclusively, associatedwith DS.

In particular, *VEGF* gene variability may be important for several angiogenesis-associated diseases, such as tumors or coronary disease [56]. Hyper-activation of the VEGFA pathway and an excess of deleterious variants in VEGFA and VEGFA pathway genes have been found to be associated with ASDs and TF [23,24,57]. Our major findings on *VEGFA* rs3025039 indicate that thehomozygous C genotype frequency is not significantly different among patients with DS affected or not by cardiac pathologies. In addition, T allele frequency is significantly lower in patients with CHDs and DS. The minor allele (T) of the rs3025039 polymorphism located inthe 3′UTR down-strand the coding region seems to be associated with lower plasma VEGF concentrations both in normal and in diseased subjects [58,59,60]. However, regardless of the effects on cytokine production, gene expression studies confirmed that homeostasis genes involved in VEGFA signaling are important for normal heart septation and they are altered in either DS or euploid patients with CHDs [61]. Our findings might therefore suggest that a genetically determined reduction of *VEGFA* production might be protective against CHDs in DS individuals.

## 5. Conclusions

Our study proposes several candidate risk loci for CHDs in DS. It is reasonable that many genes or other functional genomic elements may contribute to the development of CHDs, since a very large number of genes and signaling pathways regulate heart development, and variation in all of these genes may have a contribution (minor or major) to the risk of CHDs in DS. Our results support the view of acomplex architecture of risk alleles with cumulative effects and complex genetic and epigenetic interactions, including polymorphisms in VEGFA functional pathways [23,24], that explain the variability of DS phenotypes, in particular concerning CHDs.

## Figures and Tables

**Table 1 genes-11-01428-t001:** Baseline characteristics of 102 adult young Down syndrome (DS) patients and 85 healthy controls.

Variables	Down Syndrome Patients		Healthy Controls		*p* *
Demographic and clinical characteristics	Nr	%	Nr	%	
Age, mean ± SD	25.01 ± 4.4		27.8 ± 13.8		0.077
Female	52	51.98	53	55.21	0.31
Cardiac pathologies (DC)	59	57.84	0	-	-
No cardiac pathologies (NoC)	43	42.16	96	100	-
Congenital cardiac pathologies (CC)	39	38.23	-	-	-
Acquired cardiac pathologies after congenital lesions (ACC)	29	28.43	-	-	-
Acquired without previous congenital cardiac pathologies (AwoCC)	20	19.61	-	-	-
**Type of Congenital cardiac pathologies (*N* = 39)**					
Interatrial or interventricular defects	32	82.05			
Valvular malformations	5	12.82			
Transposition of the great arteries orTetralogy of Fallot TOF	2	5.13			
**Type of Acquired cardiac pathologies (*N* = 49)**					
Interatrial or interventricular residual defects	3	6.12			
Electric and valvular disfunctions	4	8.16			
Electric disorders	7	14.29			
Valvular disfunctions	35	71.43			

**Table 2 genes-11-01428-t002:** Genes and single nucleotide polymorphisms (SNPs) (accession number) investigated in the study.

Gene	Chromosome Localization (GRCh38/hg38)	SNP	Alleles	Effects
*INFAR2*	HSA21: 33241945	rs2229207	T>C	F8S substitution
*INFGR2*	HSA21: 33420603	rs2834213	A>G	Intron 2 variant
*IL10RB*	HSA21: 33268483	rs2834167	A>G	K47Esubstitution
*VEGFA*	HSA06: 43784799	rs3025039	C>T	3′UTRvariant

**Table 3 genes-11-01428-t003:** *IFNAR2* rs2229207 genotype frequencies of Down syndrome (DS) patient and control groups.

*IFNAR2* rs2229207		TTT		TCC/CTT		CCC	
	N.		Freq.		Freq.		Freq.
Down syndrome *	102	78	0.764	23	0.225	1	0.009
Down syndrome with cardiac pathologies (DC)	59	44	0.746	15	0.254	0	0.000
Congenital cardiac pathologies (CC)	39	29	0.744	10	0.256	0	0.000
Acquired cardiac pathologies (AC)	49	36	0.735	13	0.265	0	0.000
Acquired and congenital cardiac pathologies (ACC)	29	21	0.724	8	0.276	0	0.000
Acquired withoutcongenital cardiac pathologies (AwoCC)	20	15	0.750	5	0.250	0	0.000
Down syndrome with no cardiac pathologies (NoC)	43	34	0.791	8	0.186	1	0.023
Control subjects (CTRL)	96	79	0.823	17	0.177	0	0.000

* In Down syndrome patients, homozygous T (three copies of the T allele) or C (three copies of the C allele) were tagged, respectively, as TT and CC. Both DS and control group genotype distribution respected the Hardy–Weinberg equilibrium.

**Table 4 genes-11-01428-t004:** Statistical analyses of differences in genotype frequencies between patients with DS and control groups.

*IFNAR2* rs2229207	TTT			TCC/CTT			CCC		
	OR	95% CI	*p*	OR	95% CI	*p*	OR	95% CI	*p*
Down syndrome vs. CTRL	0.694	0.338–1.433	0.369	0.468	0.657–2.808	0.467	-	--	-
DC vs. NoC	0.776	0.303–1.988	0.643	1.491	0.567–3.920	0.478	-	--	-
CC vs. CTRL	0.6214	0.250–1.544	0.339	1.609	0.648–3.998	0.339	-	--	-
ACC vs. CTRL	0.562	0.2096–1.51	0.287	1.778	0.662–4.771	0.287	-	--	-
AwoCC vs. CTRL	0.643	0.202–2.042	0.528	1.556	0.490–4.942	0.298	-	--	-
CC vs. AwoCC	0.9667	0.279–3.346	1.000	1.034	0.299–3.581	1.000	-	--	-
AAC vs. AwoCC	0.8750	0.239–3.209	1.000	1.143	0.312–4.191	1.000	-	--	-

Differences in distribution of genotypes between patients and controls as well as among patient subgroups were analyzed by Fisher’s exact test.

**Table 5 genes-11-01428-t005:** *IFNGR2* rs2834213 genotype frequencies of patient with Down syndrome (DS) and control groups.

*IFNGR2* rs2834213	N.	AAA		GAA/GGA		GGG	
		N.	Freq.	N.	Freq.	N.	Freq.
Down syndrome *	102	39	0.382	31	0.304	32	0.314
Down with cardiac pathologies (DC)	59	28	0.475	14	0.237	17	0.288
Congenital cardiac pathologies (CC)	39	17	0.435	7	0.179	15	0.385
Acquired cardiac pathologies (AC)	49	22	0.449	12	0.245	15	0.306
Acquired and congenital cardiac pathologies (ACC)	29	11	0.3793	5	0.1724	13	0.448
Acquired withoutcongenital cardiac pathologies (AwoCC)	20	11	0.550	7	0.350	2	0.100
Down syndrome with no cardiac pathologies (NoC)	43	11	0.256	17	0.395	15	0.349
Control subjects (CTRL)	96	42	0.438	46	0.479	8	0.083

* Control group but not DS patient genotype distribution respected the Hardy–Weinberg equilibrium (DS expected frequencies: AA 29.1; AG 50.8; GG 22.1).

**Table 6 genes-11-01428-t006:** Statistical analyses of differences in genotype frequencies between Down syndrome patients and control groups.

*IFNGR2* rs2834213	AAA			GAA/GGA			GGG			A/*		
	OR	95% CI	*p*	OR	95% CI	*p*	OR	95% CI	*p*	OR	95% CI	*p*
Down syndrome vs. CTRL	0.803	0.447–1.443	0.555	0.468	0.257–0.853	0.0157	5.094	2.114–12.27	0.0001	0.196	0.081–0.473	0.0001
DC vs. NoC	2.628	1.126–6.177	0.0385	0.476	0.202–1.121	0.126	0.757	0.325–1.756	0.526	1.324	0.569–3.076	0.526
CC vs. CTRL	1.002	0.467–2.154	1.000	0.235	0.093–0.590	0.0014	6.964	2.543–19.07	0.0001	0.144	0.052–0.393	0.0001
ACC vs. CTRL	0.793	0.334–1.881	0.667	0.224	0.078–0.641	0.004	9.054	3.121–26.26	0.0001	0.111	0.038–0.320	0.0001
AwoCC vs. CTRL	1.586	0.595–4.225	0.456	0.578	0.210–1.591	0.3269	1.238	0.237–6.468	0.680	0.808	0.155–4.220	0.680
CC vs. AwoCC	0.632	0.214–1.871	0.426	0.406	0.119–1.390	0.198	5.625	1.145–27.79	0.0326	0.178	0.036–0.878	0.0326
AAC vs. AwoCC	0.500	0.157–1.591	0.260	0.214	0.047–0.968	0.189	7.313	1.427–37.48	0.0121	0.137	0.027–0.701	0.0121

A/*: Analysis of A-positive (homozygous and heterozygous: AAA, AAG and AGG) vs. A-negative genotypes. Differences in distribution of genotypes between patients with DS and controls as well as among patient subgroups were analyzed by Fisher’s exact test.

**Table 7 genes-11-01428-t007:** *IL10RB* rs2834167 genotype frequencies of patients with DS and controls.

*IL10RB* rs2834167	N.	AAA		GAA/GGA		GGG	
		N.	Freq.	N.	Freq.	N.	Freq.
Down syndrome *	93	30	0.322	58	0.624	5	0.054
Down syndrome with cardiac pathologies (DC)	54	18	0.333	32	0.593	4	0.074
Congenital cardiac pathologies (CC)	35	12	0.343	20	0.571	3	0.086
Acquired cardiac pathologies (AC)	44	14	0.318	27	0.614	3	0.068
Acquired withoutcongenital cardiac pathologies (AwoCC)	19	6	0.316	12	0.631	1	0.053
Acquired and congenital cardiac pathologies (ACC)	25	8	0.320	15	0.600	2	0.080
Down syndrome with no cardiac pathologies (NoC)	43	34	0.791	8	0.186	1	0.023
Control subjects (CTRL)	96	51	0.531	38	0.395	7	0.073

* Control but not Down syndrome patient group genotype distribution respected the Hardy–Weinberg (HW) equilibrium (HW DS expected frequencies: AA 37.4; AG 43.1; GG 12.4).

**Table 8 genes-11-01428-t008:** Statistical analyses of differences in genotype frequencies between patient with DS and controls.

*IL10RB* rs2834167	AAA			GAA/GGA			GGG			G/*		
	OR	95% CI	*p*	OR	95% CI	*p*	OR	95% CI	*p*	OR	95% CI	*p*
Down syndrome vs. CTRL	0.402	0.218–0.742	0.0038	2.641	1.436–4.858	0.0024	0.729	0.214–2.484	0.758	2.487	1.347–4.590	0.0038
DC vs. NoC	0.422	0.211–0.475	0.0001	3.185	1.641–6.183	0.0001	3.185	0.369–27.47	0.378	9.714	3.605–26.17	0.0001
CC vs. CTRL	0.632	0.384–1.042	0.069	1.482	0.998–2.200	0.071	1.186	0.314–4.478	1.000	0.778	0.394–1.537	0.069
ACC vs. CTRL	0.397	0.155–1.022	0.0682	2.391	0.958–5.965	0.0686	1.116	0.211–5.911	1.000	2.516	0.978–6.474	0.0682
AwoCC vs. CTRL	0.390	0.135–1.124	0.126	2.732	0.974–7.660	0.072	0.713	0.081–6.300	1.000	2.566	0.889–7.403	0.126
CC vs. AwoCC	1.086	0.485–2.430	1.000	0.905	0.578–1.416	0.7753	1.629	0.182–14.61	1.000	0.885	0.268–2.917	1.000
AAC vs. AwoCC	1.013	0.423–2.430	1.000	0.950	0.594–1.519	1.000	1.520	0.148–15.56	1.000	0.981	0.272–3.533	1.000

G/*: Analysis of G-positive (homozygous and heterozygous: GGG, AAG and AGG) vs. G-negative genotypes. Differences in distribution of genotypes between patients with DS and controls as well as among patient subgroups were analyzed by Fisher’s exact test.

**Table 9 genes-11-01428-t009:** *VEGFA* rs3025039 genotype frequencies of patients with DS and controls.

*VEGFA* rs3025039	N.	CC		CT		TT	
		N.	Freq.	N.	Freq.	N	Freq.
Down syndrome	102	83	0.814	17	0.166	2	0.020
Down syndrome with cardiac pathologies (DC)	59	47	0.797	11	0.186	1	0.017
Congenital cardiac pathologies (CC)	39	34	0.872	4	0.102	1	0.026
Acquired cardiac pathologies (AC)	49	38	0.776	10	0.204	1	0.020
Acquired and congenital cardiac pathologies (ACC)	29	25	0.862	3	0.103	1	0.035
Acquired withoutcongenital cardiac pathologies (AwoCC)	20	13	0.650	7	0.350	0	0000
Down syndrome with no cardiac pathologies (NoC)	43	36	0.837	6	0.139	1	0.023
Control subjects (CTRL)	96	62	0.645	30	0.313	4	0.042

**Table 10 genes-11-01428-t010:** Statistical analyses of differences in genotype frequencies between patient and control groups.

*VEGFA* rs3025039	CC			CT			TT			T/*		
	OR	95% CI	*p*	OR	95% CI	*p*	OR	95% CI	*p*	OR	95% CI	*p*
Down syndrome vs. CTRL	2.262	1.157–4.424	0.019	0.480	0.239 0.966	0.052	0.405	0.072–2.268	0.413	0.442	0.226–0.864	0.019
DC vs. NoC	0.762	0.272–2.130	0.797	1.413	0.478–4.176	0.599	0.724	0.044–11.92	1.000	1.313	0.469–3.673	0.797
CC vs. CTRL	3.521	1.244–9.969	0.017	0.274	0.088–0.853	0.022	0.533	0.058–4.934	1.000	0.284	0.100–0.804	0.017
ACC vs. CTRL	3.237	1.028–10.19	0.0562	0.277	0.077–0.999	0.0465	0.723	0.078–6.750	1.000	0.309	0.098–0.973	0.056
AwoCC vs. CTRL	0.962	0.346–2.674	1.000	1.292	0.461–3.623	0.602	0.442	0.023–8.545	1.000	1.040	0.374–2.891	1.000
CC vs. AwoCC	3.662	0.984–13.62	0.084	0.212	0.053–0.847	0.033	1.783	0.069–45.87	1.000	0.273	0.073–1.016	0.084
AAC vs. AwoCC	3.365	0.830–13.64	0.097	0.214	0.047–0.967	0.068	2.158	0.084–55.73	1.000	0.297	0.073–1.205	0.097

Both Down syndrome and control group genotype distribution respected the Hardy–Weinberg equilibrium. T/*: Analysis of T-positive (homozygous and heterozygous: TT and CT) vs.T-negative genotypes. Differences in distribution of genotypes between DS patients and control as well as among patient subgroups were analyzed by Fisher’s exact test.

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
