# Peer review of "Susceptibility to Heart Defects in Down Syndrome Is Associated with Single Nucleotide Polymorphisms in HAS 21 Interferon Receptor Cluster and VEGFA Genes"

_genes, 2020, doi:10.3390/genes11121428_

Round 1

Reviewer 1 Report

The manuscript by Balistreri et al. suggests an assocation between an interferon cluster of genes and VEGFA genes with a higher susceptibility to heart defects in Down syndrome. 

The introduction is well written and provides all the necessary information in order to understand the reasoning behind the study. I would recommend some language editing and also in lines 82-84 some references would be helpful. The authors mention:

"On the other hand in animal models the increased dosage of genes on the homologue of HSA21, explains only in part of?? the increased risk to CHD and suggests the involvement of other genes located on other chromosomes." Some references at the end of this paragraph would be helpful.

The methods are well described and similar to the introduction, some minor editing would be advisable.

The results section is difficult to understand. First of all, the same format should be maintained when authors are describing the figures in the text. For example, for figures 3 and 4, authors don't separate between 3a and 3b or 4a and 4b respectively but in the case of figures 5 and 6 they do. Also, describing each time in the text what figure they are refering to would be helpful.

Additionally, some results would benefit from a more detail and accurate description. For example, when talking about the VEGFA genotype distributions, authors mention that the CC genotype..."is well represented in Down patients with congenital cardiac pathologies. These patients, had a significant lower frequency of CT genotype when compared to patients with heart defects developed after birth (AwoCC)." Looking at the table 6a we see that the CC genotype frequency among Down syndrome patients with congenital cardiopathies is 0.872 and that their CT genotype had a frequency of 0.102 compared to a CT frequency of 0.350 in AwoCC patients. While this is true it would be interesting to mention that the CC genotype among Down syndrome patients with no cardiopathies has a frequency of 0.837. This shows that Down syndrome patients with congenital cardiopathies and Down syndrome patients with no cardiopathies had very similar CC frequencies.

I would recommend rewritting the results section in a more detail and accurate way in order to better understand the results.

The discussion section should be modified accordingly.

Reviewer 2 Report

The authors describe variants that are potentially related to the increased risk of heart defects in individuals with Down syndrome.

Comments for the authors:

You sought to identify some candidate risk SNPs for heart defects and acquired heart conditions in individuals with Down syndrome. I had difficulty understanding how these variants would lead to heart defects in Down syndrome especially. The questions that I had were:

  1. Is it the additional dosage of that SNP that leads to increased risk for heart defects? Would having 3 doses place a child with Down syndrome at the greatest risk?
  2. If only 2 copies of that variant are sufficient, then have you considered examining children with euploidy and CHDs to determine if they also have homozygosity for the risk variants.
  3. Do you hypothesize that these particular risk variants impact the expression of other genes found in triplicate, thus leading to CHD?

You may have tried to explain that, but if so, it didn’t seem to come across very clearly.

A couple of other points: first, line 18 – the convention is to not refer to individuals as “DS patients” or “DS cases” but rather patients (or individuals) with Down syndrome; second, line 36 – it is not Down’s syndrome but Down syndrome. Both of these are on page 1.

Overall, this was an interesting study, but I’m not sure the impact of these findings was made clear.

Round 2

Reviewer 1 Report

Thanks for the changes. I recommend some minor english language editing.

Author Response

English language of the manuscript was checked and editing errors corrected according to kind referee suggestions

Reviewer 2 Report

The paper is more clear to me know. However, I still find it confusing to indicate genotypes of alleles present in 3 copies as GG, etc. Could you simply put the full genotypes you are investigating in the table and discussion?  It might make it easier for the reader to understand your study and its results. 

Author Response

According to referee suggestions, captions, column headings and notes of tables 3a (lines 205-207), 3b (line 208), 4a (lines 210 to 212), 4b (lines 215-216), 5a (lines 219-221) and 5b (lines 224-225) were changed.

English language of the manuscript was carefully checked and editing errors corrected